# Beyond Candidates
## : Adaptive Dialogue Agent Utilizing Persona and Knowledge

**Jungwoo Lim**[*], **Myunghoon Kang**[*], **Jinsung Kim**[*], **Jeongwook Kim**,
**Yuna Hur, Heuiseok Lim**[†]

Department of Computer Science and Engineering, Korea University

{wjddn803,chaos8527,jin62304,k0s1k0s1k0,yj72722,limhseok}@korea.ac.kr

## Abstract

To build ultimate dialogue agents, previous studies suggest models that ground both persona and knowledge. However, applying the dialogue system directly to the usual conversation is still limited because the system requires a complete sentence-formed persona and knowledge candidate sets from the given dataset. In contrast to the dialogue setting in the dataset, humans utilize semantic concepts in their minds rather than a set of pre-defined candidate sentences. Following this manner of human dialogue, we suggest an adaptive dialogue system that is applicable to situations where complete sentence-formed candidates are not given. Our model generates consistent and relevant persona descriptions and identifies relevant knowledge for engaging and knowledgeable responses, even with fragmentary information. We show that our model outperforms previous baselines that utilize persona and knowledge candidate sentences and conduct the human evaluation on the machine-generated responses. In addition, we conduct ablation studies to demonstrate the effectiveness of each component of our model. Furthermore, we apply our model to other dialogue datasets that only ground knowledge or persona to showcase its adaptability. Our code is available at https://github.com/dlawjddn803/BeCand.

## 1 Introduction

In usual conversations, humans utilize the semantic concept in their minds in terms of the dialogue topic and the preference of the interlocutor. With the semantic-level of concepts, humans communicate each other by aggregating the concepts to convey knowledgeable and empathetic responses (Collins and Quillian, 1969). It implies that people converse by adaptively reorganizing and retrieving additional information with their

---

[*]These authors contributed equally to this work.
[†]Corresponding author.

Figure 1: Comparison of conversational settings. (a) is a candidate-free conversational setting and the machine's answer in (a) is a generated response from our model. (b) is a previous conversational setting and the machine's answer in (b) is the response from the BART-large model trained on FoCus dataset.

semantic concepts, encompassing knowledge and persona, not by relying on pre-defined sources (Young et al., 2018; Zou et al., 2021; Li et al., 2023).

It seems that Jang et al. (2022a) and Lim et al. (2022) adhere to this human-like approach

on the conversation by referring to persona and knowledge. However, it neglects the humans' semantic concept reconstruction and retrieval capability by requiring pre-defined candidate sets to ground as in Figure 1 (b). As knowledge and persona candidates for the agents are not given in usual conversation, the dependency on the candidates eventually limits their applicability to candidate-free situations as depicted in Figure 1 (a).

To build the dialogue agents adaptive to the candidate-agnostic situation, two branches of studies are conducted. In knowledge-grounded conversation, the knowledgeable agents employ the non-parametric memory-based retrieval to overcome candidate-agnostic situations (Lewis et al., 2020b; Paranjape et al.). Similarly, persona-aware dialogue agents consider the out-of-persona situations by extending persona sentences from a few persona concept (Xu et al., 2020; Liu et al., 2022; Li et al., 2023). Even though both streams of research focus on the candidate-agnostic conversational situation, they only leverage a single source for grounding, rather than utilizing both persona and knowledge, simultaneously.

In this paper, we propose a dialogue agent utilizing persona and knowledge that is adaptive to the candidate-free situation. To this end, our method consists of 1) a knowledge-retriever 2) a concept-based persona generator, 3) a dialogue-persona aligner, and 4) a response generator. When the knowledge concept is given, a knowledge retriever finds the relevant knowledge from the knowledge base. Our concept-based persona generator then produces complete sentences with fragmentary persona concepts. The generated persona descriptions are then validated based on the persona aligner regarding both consistency and relevancy. The validated persona descriptions are used as the input of the response generator.

Experimental results show that our candidate-free model outperforms other baselines. Also, we show that the concept-based persona generator and persona aligner boost the performance of the dialogue agents with the ablation studies. We conduct the human evaluation of our model's responses, and the result implies that our method is effective in building a persona-knowledge dialogue agent without candidate sentences. Moreover, we demonstrate that our method is capable of utilizing other dialogue datasets grounding single source,

such as PersonaChat (Zhang et al., 2018) or Wizard-of-Wikipedia (WoW) (Dinan et al., 2018), and shows the adaptiveness of our proposed model. In qualitative results, it is shown that the generated responses are comparable to the ground truth answers without the given candidates.

## 2 Related Works

### 2.1 Knowledge-grounded Dialogue System

For the informative dialogue generation, Dinan et al. (2018) and Zhou et al. (2018) introduce open-domain dialogue datasets. Various works directly exploit external knowledge to obtain informative responses (Karpukhin et al., 2020; Lee et al., 2021; Wu et al., 2022a) in knowledge-grounded conversation. Other studies exploit augmenting knowledge base to the language model with non-parametric memory-based retriever (Lewis et al., 2020b; Guu et al., 2020; Izacard and Grave, 2021). It is found that a retrieval-augmented generator also reduces hallucination in knowledge-grounded conversation as well (Shuster et al., 2021), and a similar approach recently achieves comparable performance in knowledge-grounded conversation (Paranjape et al., 2021).

### 2.2 Persona-grounded Dialogue System

Also, persona-concentrated datasets have been proposed for constructing persona-engaging dialogue agents (Zhang et al., 2018; Rashkin et al., 2019; Dinan et al., 2020; Smith et al., 2020). While Song et al. (2020) and Wu et al. (2021) focus on injecting persona with utterance post-editing, Zheng et al. (2020) devises the attention routing mechanism for handling persona dialogue. Furthermore, another research takes into account the consistency and relevancy of the persona by employing natural language inference-based critic with a consistency score in reinforcement learning. Moreover, to maintain a consistent persona perceived by the dialogue agent, Bae et al. (2022) use iterative feedback between pre-trained language models (PLMs) and human annotators.

Along with the previous studies, research that attempts to expand the persona sentences to cover candidate-free conversational settings has appeared. In other words, when given fragmentary information on the user's persona, research to complete the insufficient information is conducted using retrieval (Liu et al., 2022; Majumder et al.,

2021; Han et al., 2022) or generation (Zhou et al., 2021; Lu et al., 2022). In addition, the persona extension approach leveraging commonsense is introduced. Majumder et al. (2020) expands the given persona sentences by fetching additional information from a commonsense knowledge graph.

### 2.3 Persona and Knowledge Grounded Dialogue System

In recent studies, there has been research on fusing persona and knowledge to generate engaging and knowledgeable responses. Fu et al. (2022) suggest a persona memory in knowledge selection for persona-consistent response generation. In addition, Jang et al. (2022b) and Lim et al. (2022) propose a model that explicitly grounds both persona and knowledge simultaneously. Along with the studies, Wu et al. (2022b) ground persona, knowledge, and commonsense by fusing separate encoder-decoder structures. Furthermore, Blenderbot 3 (Shuster et al., 2022) models the persona as the long-term memory of both users and chatbot. However, none of these systems cannot be adopted to the candidate-free conversation setting due to the dependency on the given sentence-formed candidates.

### 3 Method

We propose adaptive dialogue agents that generate the responses without the persona and knowledge candidates. To this end, we assume that the knowledge and persona concepts are only given to the agent for knowledgeable and engaging responses. First, **1) knowledge retriever** retrieves the relevant paragraphs with the knowledge concept, and **2) concept-based persona generator** produces the persona descriptions with the given short persona concepts. Then, **3) persona aligner** decides whether the generated persona descriptions are relevant to the dialogue history and whether the sentences are consistent with the previous dialogue history. Afterward, **4) response generator** provides knowledgeable and engaging responses with the predicted knowledge paragraphs and persona descriptions.

### 3.1 Notation

The given dialogue $D$ is notated as $\{(u_1^{hm}, u_1^{mc}), ...(u_n^{hm}, u_n^{mc})\}$ and where $n$ is the number of rounds. $u^{hm}$ and $u^{mc}$ denote

the utterances of human and machines, respectively. The dialogue history $H$ is $\{(u_{n-w}^{hm}, u_{n-w}^{mc}), ..., (u_{n-1}^{hm}, u_{n-1}^{mc}), (u_n^{hm})\}$ where $w$ is the window size. The set of given persona sentences $P = \{p_1, p_2...p_{|P|}\}$ and $|P|$ is the number of persona sentences. Also, $C^P = \{c_1^p, c_2^p, ...c_{|P|}^p\}$ indicates the persona concepts whereas $C^K$ is a knowledge concept which is a title of the knowledge.

### 3.2 Knowledge Retriever

To let the model adapt to the situation where the knowledge candidates are absent, we use a non-parametric memory-based retrieval. We combine the query encoder and dense vector index, which is obtained from a pre-trained dense passage retriever (DPR) (Karpukhin et al., 2020) for enhanced semantic search. The retriever refers to the knowledge index from the Wikipedia knowledge which is leveraged with FAISS (Johnson et al., 2019) library. Therefore, our retriever $R(\cdot)$ finds the relevant knowledge from the index with the knowledge concept $C^K$ by using maximum inner-product search (MIPS) following Lewis et al. (2020b). The predicted top-k relevant paragraphs are then used as the input for the model and denoted as $\hat{K}$.

$$R(\hat{K}|C^K) \propto exp(\mathbf{e}(\hat{K})^\top \mathbf{q}(C^K)), \quad (1)$$

where $\mathbf{e}(\cdot)$ is an embedding from a context encoder, and $\mathbf{q}(\cdot)$ is a representation from a query encoder, both implemented with BERT (Kenton and Toutanova, 2019) pre-trained on natural-question dataset (Kwiatkowski et al., 2019).

### 3.3 Concept-based Persona Generator

To let the model exploit the semantic concept from the candidate-free situation, we propose a concept-based persona generator to provide complete persona descriptions only with the persona concepts. In detail, our persona generator is pre-trained to generate plausible full persona descriptions with only persona concepts in a retrieve-and-generate manner, following Hashimoto et al. (2018). Then, we freeze the persona generator for the response generation.

For the pre-training process, we first build the persona pool with the collections of unique persona sentences from FoCus (Jang et al., 2022a) and PersonaChat (Zhang et al., 2018).

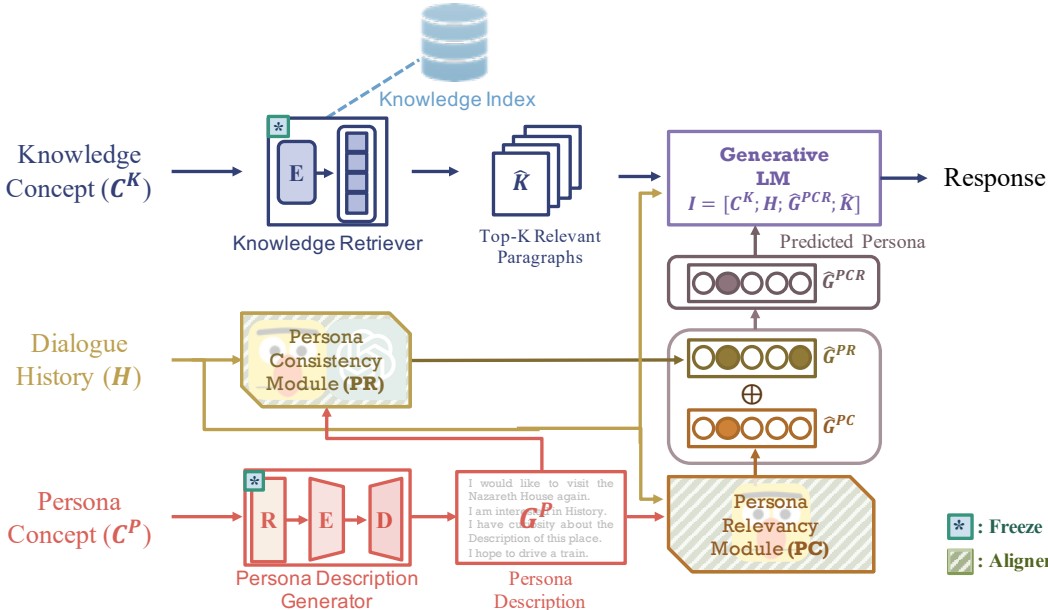

Figure 2: Overall architecture of our model.

Then, we pre-train the persona retriever using DPR (Karpukhin et al., 2020) and regard highly ranked persona sentences from BM25 (Robertson and Zaragoza, 2009) as negative samples. We then train the generator by considering top $k$ relevant persona sentences $P'_k = \{p'_1, p'_2, ..., p'_k\}$ as positive samples with BART (Lewis et al., 2020a). Our concept-based persona generator provides complete persona sentences $G^P$. The training details are presented in Appendix A.

### 3.4 Persona Aligner

The persona aligner consists of two modules, i.e., the persona consistency (PC) module and the persona relevancy module (PR). When the generated persona sentences $G^P$ are obtained, the persona consistency module predicts whether the generated persona sentences contradict the previous dialogue history $H$. However, collecting the labels of generated personas' consistency is time-consuming and labor-intensive. Therefore, we distill ChatGPT (OpenAI-Blog, 2022) as model annotators with the BERT-base model (Kenton and Toutanova, 2019) inspired by the high reasoning capability on natural language inference of the ChatGPT (Laskar et al., 2023). We asked the ChatGPT to predict whether the single persona sentence contradicts the given dialogue or not. The prompt for an alignment check is illustrated in Appendix 9. Then, the consistency module trains on the label that ChatGPT provided in a binary classification manner. The trained persona consistency module is then frozen and predicts whether the sentence is consistent with the dialogue history in the inference stage.

$$\hat{G}_i^{PC} = \text{PC}([H; G_i^P]) \qquad (2)$$

Different from the persona consistency module, the persona relevancy module takes charge of the relevancy by selecting proper persona sentences that are relevant to dialogue. Even though the persona descriptions do not conflict with the dialogue history, it is still unrevealed the level of relevancy of the persona sentence. For enhanced relevancy prediction, we first separately encode the dialogue and generated persona sentences with the question encoder with DPR, and obtain each hidden state from the last layer. Then, we concatenate the embeddings and pass them into the two linear layers to predict the relevancy of the persona sentences to the dialogue.

$$\hat{G}_i^{PR} = \text{PR}([H; G_i^P]) \qquad (3)$$

If the two modules both predict the sentence as relevant and consistent, we assume the sentences are aligned with the given dialogue.

$$\hat{G}_i^{PCR} = \{\hat{G}_i^{PC} \cap \hat{G}_i^{PR}\}, \qquad (4)$$

We compute the loss as Equation 5.

$$\mathcal{L}_P = \\ -\sum_j l_j \cdot \log \hat{l}_j + (1 - l_j) \cdot \log(1 - \hat{l}_j) \qquad (5)$$

| Model | Backbone | Candidate Usage | BLEU | chrF++ | R-1 | R-2 | R-L |
|---|---|---|---|---|---|---|---|
| *Baselines (w/ Candidates)* | | | | | | | |
| Jang et al. (2022a) | GPT2-small | O | 11.43 | 28.73 | 36.58 | 19.44 | 32.62 |
| | GPT2-medium | O | 12.31 | 30.12 | 38.29 | 21.17 | 34.12 |
| | BART-base | O | 11.99 | 29.77 | 36.24 | 19.73 | 32.13 |
| | BART-large | O | 11.91 | 30.69 | 36.57 | 19.83 | 32.05 |
| INFO (Lim et al. (2022)) | RAG | O | **31.46** | **53.29** | **58.26** | **42.35** | **53.06** |
| *w/o Candidates* | | | | | | | |
| Lewis et al. (2020a) | BART-large | X | 13.14 | 31.45 | 38.67 | 19.93 | 34.13 |
| Lewis et al. (2020b) | RAG | X | 15.90 | 35.50 | 41.21 | 22.80 | 36.45 |
| Ours | BART-large | X | 20.86 | 40.12 | 45.89 | 28.06 | 40.97 |
| | RAG | X | 20.30 | 39.53 | 45.17 | 28.05 | 40.51 |

Table 1: Focus Results. Main results on the official validation set. The models are evaluated by generation metrics, including BLEU, chrF++, ROUGE-1 (R-1), ROUGE-2 (R-2), ROUGE-L (R-L).

Note that $l_j$ is the ground-truth label of the $j$-th example.

### 3.5 Response Generator

With the predicted relevant knowledge passages and persona descriptions, we concatenate them into one sequence along with the concept of the knowledge and dialogue as $I = [C^K; H; \hat{G}^{PCR}; \hat{K}]$. Then, we pass into the generative language model to obtain the responses, and the language modeling loss is computed as Equation 6.

$$\mathcal{L}_{LM} = -\sum_{i=1}^{T} \log \text{Prob}(t_i|t_1,...,t_{i-1}), \quad (6)$$

where $\text{Prob}(\cdot)$ denotes a probability of the generative langauge model, $t_i$ is $i$-th token of target sentence, and $T$ is the number of tokens. The final loss function $\mathcal{L}_{Final}$ is computed as Equation 7 and $\lambda_P$ and $\lambda_{LM}$ are hyperparameters.

$$\mathcal{L}_{Final} = \lambda_P \mathcal{L}_P + \lambda_{LM} \mathcal{L}_{LM} \quad (7)$$

## 4 Experimental Setup

### 4.1 Dataset

**FoCus** (Jang et al., 2022a) is a dataset designed for dialogue models that utilize knowledge and persona simultaneously. It comprises 12,484 dialogues, 5,152 Wikipedia knowledge, and 32,855 persona sentences. We conduct the experiments on the official FoCus validation set since the official test set can only be confirmed via the leaderboard on the Codalab platform[1].

[1] https://codalab.lisn.upsaclay.fr/competitions/3754

**Wizard of Wikipedia (WoW)** WoW (Dinan et al., 2018) contains 22,311 dialogues with 201,999 turns that utilize Wikipedia articles, primarily aimed at facilitating knowledge-based dialogue. In WoW datases, the wizard responds to the apprentice based on selected knowledge. We utilized the test splits for Wizard of Wikipedia (WoW) for the experiments.

**PersonaChat** PersonaChat (Zhang et al., 2018) consists of 162,064 utterances between two interlocutors, each with 3-5 different persona sentences. The interlocutors converse with their persona, attempting to figure out the persona of others. We used the revised validation set for PersonaChat experiments.

### 4.2 Baselines

**BART** (Lewis et al., 2020a) is a language model that integrates concepts from both autoregressive and denoising autoencoder models, making it highly effective for text generation. For the experiments, we only use the BART$_{large}$ model.

**RAG** Retrieval-augmented generator (RAG) (Lewis et al., 2020b) is a model which combines pre-trained parametric and non-parametric memory for language generation. It shows comparable performances on open-domain question-answering and conversation.

**BART+PG+KG** Jang et al. (2022a) introduce the dialogue system that can provide personalized responses by considering the interlocutor's persona and external knowledge. The model finds relevant paragraphs by considering the last utterance and selects proper sources from the persona (PG) and

| Dataset | Model | Backbone | Info. ↑ | Hal. (K) ↓ | Hal. (P) ↓ | Rel. (P) ↑ | Con. (P) ↑ | Flu. ↑ |
|---------|-------|----------|---------|-----------|-----------|-----------|-----------|--------|
| FoCus | Lewis et al. (2020a) | BART-large | 2.33 | 1.97 | 1.57 | 2.06 | 2.11 | 2.55 |
| | Ours | | 2.42 | 1.74 | 1.44 | 2.32 | 2.39 | 2.61 |
| | Lewis et al. (2020b) | RAG | 2.15 | 1.98 | 1.46 | 1.86 | 1.99 | 2.29 |
| | Ours | | 2.52 | 1.64 | 1.38 | 2.48 | 2.46 | 2.60 |

Table 2: Human evaluation results. Info.: informativeness, Hal.(K): hallucination in knowledge and Hal.(P): hallucination in persona, Rel. (P): relevancy of persona usage, Con. (P): consistency of persona usage, Flu.: fluency.

knowledge candidates (KG). After obtaining useful sources, the model generates the responses in an auto-regressive manner.

**INFO** Lim et al. (2022) propose a conversational model that grounds both persona and knowledge. The model consists of a knowledge selector, a persona selector, and a responses generator. Each selector is implemented with poly-encoder (Humeau et al., 2020) and the output is used as the query for the response generator based on RAG (Lewis et al., 2020b).

### 4.3 Evaluation Metrics

The official automatic evaluation metrics for the FoCus benchmark include BLEU (Papineni et al., 2002), chrF++ (Popović, 2017), ROUGE-1, ROUGE-2, and ROUGE-L (Lin, 2004). These metrics are frequently employed to compare machine-generated responses to gold responses in generation tasks. In our experiments on PersonaChat, we also report the performance of the unigram F1 metrics.

For human evaluation, we adopt six metrics on response generation. 1) **Informativeness** measures the extent of the information conveyed within a response and denotes the degree of providing new, valuable, or relevant details, insights, or facts to the conversation. We also have two criteria for hallucination regarding persona and knowledge. 2) **Knowledge hallucination** is the metric that shows the level of hallucination of generated output that contradicts reality. Similarly, 3) **Persona hallucination** is the metric that indicates the hallucination level based on the given persona descriptions. Along with the persona-related metrics, 4) **Persona relevancy** metric denotes how much the given persona directly relates to the ongoing conversation. Moreover, 5) **Persona consistency** refers to how consistently the persona is maintained in a given dialogue. Lastly, 6) **Fluency** measures the ability to communicate

smoothly, effortlessly, and coherently. Details of our experiments are provided in Appendix B.

## 5 Results and Discussion

### 5.1 Automatic Evaluation

We conduct the experiments on FoCus dataset to show our method's effectiveness without the given candidates. Table 1 demonstrates that our method achieves the second-highest score while the first-ranked model directly exploits the persona and knowledge candidate set. In addition, our method outperforms the performance of Jang et al. (2022a) even though it utilizes the candidates from the dataset. Furthermore, all models incorporating our method outperform their vanilla backbone models significantly.

### 5.2 Human Evaluation

We also demonstrate the effectiveness of our method through human evaluations. We recruit nine human workers who have at least bachelor's degree and are proficient in English. We randomly chose 30 dialogues from each datasets. We asked the workers to evaluate the machine-generated responses according to six criteria described earlier. The score is scaled from 1 to 3, and the results are indicated in Table 2. The results indicate that our method is effective in achieving both persona consistency and persona relevancy. Moreover, our method shows comparable performance in decreasing both knowledge hallucination and persona hallucination in the FoCus dataset.

### 5.3 Ablation Studies

We also conduct ablation studies on our methods with respect to the knowledge retriever, concept-based persona generator, and persona aligner.

**Knowledge Retriever** To demonstrate the effectiveness of our knowledge retriever in our model, we compare vanilla backbone which is fine-tuned on the FoCus dataset, and our model.

As shown in Table 3, incorporating relevant knowledge into the input of the vanilla generative language models enhances the performance of the response generation regardless of the backbone language models. Also, the knowledge retriever enhances the performances consistently, even when the persona generator and aligner are combined in our method. The performance decrease of the models without the knowledge retriever[2] suggests that our knowledge retriever is effective.

| Model | Backbone | K-Retr. | BLEU | chrF++ | R-1 | R-2 | R-L |
|---|---|---|---|---|---|---|---|
| Lewis et al. (2020a) | BART | X | 13.14 | 31.45 | 38.67 | 19.93 | 34.13 |
| | | O | 14.83 | 33.86 | 39.90 | 21.07 | 34.95 |
| Lewis et al. (2020b) | RAG | X | 15.90 | 35.50 | 41.21 | 22.80 | 36.45 |
| | | O | 14.66 | 33.90 | 38.79 | 20.42 | 33.87 |
| Ours | BART | X | 12.66 | 34.22 | 38.73 | 19.69 | 32.96 |
| | | O | **20.86** | **40.12** | **45.89** | **28.06** | **40.97** |
| | RAG | X | 19.46 | 38.21 | 43.58 | 25.98 | 38.87 |
| | | O | **20.30** | **39.53** | **45.17** | **28.05** | **40.51** |

Table 3: Ablation study on knowledge retriever. K-Retr. denotes the knowledge retriever.

**Persona Generator** We also compare the performance of models by ablating the type of persona descriptions. "GT" refers to our model utilizing ground-truth persona sentences, while "random" indicates the models with five random persona descriptions from the persona pool. As shown in Table 4, the models in random settings exhibit a decrease in performance, regardless of the backbone models. However, the proposed method based on the RAG model outperforms the model that utilizes ground-truth persona descriptions. This suggests that the generated persona descriptions from our concept-based persona generator are comparable to the ground-truth persona sentences and that our concept-based persona generator can replace the labor-intensive human annotating process.

| Model | Backbone | Persona Desc. | BLEU | chrF++ | R-1 | R-2 | R-L |
|---|---|---|---|---|---|---|---|
| Ours | BART | GT. | **20.86** | **40.23** | **46.31** | **28.42** | **41.37** |
| | | Random | 19.76 | 38.85 | 45.40 | 27.45 | 40.55 |
| | | Ours | **20.86** | 40.12 | 45.89 | 28.06 | 40.97 |
| | RAG | GT. | 19.55 | 38.77 | 44.65 | 27.02 | 39.9 |
| | | Random | 17.50 | 36.51 | 42.50 | 24.72 | 37.8 |
| | | Ours | **20.30** | **39.53** | **45.17** | **28.05** | **40.51** |

Table 4: Ablation study on concept-based persona generator. GT. denotes the ground truth persona sentences which are given from the dataset.

| Model | Backbone | Persona Aligner | BLEU | chrF++ | R-1 | R-2 | R-L |
|---|---|---|---|---|---|---|---|
| Ours | BART | GT. | **20.96** | **40.45** | **48.08** | **29.24** | **42.67** |
| | | Random | 3.79 | 14.79 | 19.41 | 3.94 | 16.69 |
| | | Ours | 20.86 | 40.12 | 45.89 | 28.06 | 40.97 |
| | RAG | GT. | **20.74** | **40.77** | **47.01** | **28.29** | **41.55** |
| | | Random | 17.54 | 36.81 | 42.08 | 23.89 | 37.16 |
| | | Ours | 20.30 | 39.53 | 45.17 | 28.05 | 40.51 |

Table 5: Ablation study on persona aligner. GT. indicates the ground truth label of persona selection from the dataset.

| Dataset | Model | F1 | BLEU | chrF++ | RL |
|---|---|---|---|---|---|
| WoW | RAG[†] (2020b) | - | - | - | 11.57 |
| | FiD (2021) | - | - | - | 16.06 |
| | Hindsight (2021) | - | - | - | 17.06 |
| | QKConv (2022) | - | - | - | **17.72** |
| | BART (Ours) | 17.11 | 8.31 | 18.59 | 16.35 |
| | RAG (Ours) | 16.39 | 8.07 | 18.33 | 15.91 |
| PersonaChat | KV Profile Memory (2018) | 13.65 | - | - | - |
| | TransferTransfo (2019) | 15.71 | - | - | - |
| | $\mathcal{P}^2$ Bot (2020) | 19.08 | - | - | - |
| | LMEDR (2023) | **21.99** | - | - | - |
| | BART (Ours) | 19.46 | 12.78 | 15.7 | 18.44 |
| | RAG (Ours) | 18.67 | 11.26 | 16.73 | 17.48 |

Table 6: Automatic evaluation results on the other dialogue datasets. [†] denotes the vanilla model and the scores of other models in WoW are imported from KILT benchmark (Petroni et al., 2021).

**Persona Aligner** We evaluate the impact of our consistency and relevancy module by employing the persona relevancy label. "GT" refers to utilizing the ground-truth persona relevancy label in the FoCus dataset, while "random" indicates the model with randomly assigned persona relevancy labels. As shown in Table 5, our method performs comparably to the GT performance when trained on both BART and RAG. However, performance significantly decreases in random settings. This indicates that our persona aligner effectively captures the consistency and relevancy of the generated persona with respect to the dialogue context.

## 5.4 Adaptation to Other Dialogue Datasets

To evaluate the adaptiveness of our method, we conduct experiments on the other dialogue datasets, Wizard of Wikipedia (WoW) and PersonaChat. Since both datasets consist of dialogues grounding a single source, there are no candidates for the other source. In other words, there are no persona candidates in WoW dataset, and knowledge candidates are absent in PersonaChat. Therefore, we report the results of applying our method to these single-source datasets. Table 6 demonstrates that the models with our method show comparable

---

[2]Meanwhile, we measure the retriever's sole performance by checking whether the retrieved passages belong to the ground-truth knowledge source. The accuracy of the knowledge retriever is 75.19 %.

performances in ROUGE-L. Also, our method based on BART exceeds FiD (Izacard and Grave, 2021) which shows remarkable performances in knowledge-grounded conversation. Also, our BART-based model in PersonaChat also surpasses the $\mathcal{P}^2$ Bot (Liu et al., 2020) according to the unigram-F1 metric.

## 5.5 Qualitative Results

| Knowledge Concepts |
| --- |
| Cardiff Bay Barrage |
| **Human's Persona Concepts** |
| love, area. 
 have, Wales, been 
 engineering, am, fan 
 Europe, like, visit 
 Kingdom, not, United, am |
| **Dialogue** |
| *Human*: Where is it located? 
 *Machine*: You've never been there before, but this can be found in Wales between the Queen Alexandra Dock and Penarth Head. 
 *Human*: Was this a large project? |
| **Generated Persona** |
| I love the downtown area. 
 I have never been to Wales. 
 I am a fan of engineering. 
 I would like to visit Europe. 
 I am from the United Kingdom. |

| Predicted Responses | |
| --- | --- |
| BART[‡] | Oh yes. You'd be interested in this barrage, which was around 60 million deep and was the largest civil engineering project in the world. |
| RAG[‡] | Oh yes. You'll be interested in this barrage, with its 4.75 mile long effect. It was the largest civil engineering project in Europe. |
| **Ours (BART)** | Oh yes, it was very large. You'd be a fan of this engineering work, which was considered one of the largest civil engineering projects ever undertaken in the country. |
| **Ours (RAG)** | It was very large in scale, and you'd be a fan of this engineering, as it was one of the largest civil engineering projects in the country. |

| Ground Truth Knowledge |
| --- |
| It was one of the largest civil engineering projects in Europe during construction in the 1990s. |
| **Ground Truth Persona** |
| I love the bay area. 
 I have never been to Wales. 
 I am a fan of engineering. 
 I would like to visit Europe. 
 I am not from the United Kingdom. |
| **Ground Truth Response** |
| Oh yes, very large. With you being a fan of engineering, you'd be interested to hear that this was one of the largest civil engineering projects in Europe during the time. |

Table 7: Qualitative result. [‡] denotes the vanilla models, and red and blue each indicate the parts utilized by Ours in the persona and the knowledge. All the predicted results are from our model.

Table 7 demonstrates the prediction results from the baselines and our models on the FoCus dataset. It is noteworthy that the vanilla BART and RAG models tend to generate shallow responses that revolve around the topic of Cardiff Bay Barrage. Furthermore, the models tend to provide numerical information that lacks factual support regarding the knowledge concept. To sum up, these models fail to achieve a deep understanding o human preferences based on the provided persona concepts, leading to less engaging responses that lack any persona-related expressions.

Furthermore, our proposed models generate informative and empathetic responses, striking a balance between incorporating external information relevant to the knowledge concept and avoiding any distortion. For instance, our models generate expressions such as "it was one of the largest civil engineering projects in the country," which provide sufficient information. These results suggest that our method is well-suited for scenarios where the sentence-formed knowledge and persona candidates are absent.

## 6 Conclusions

In this paper, we introduced an adaptive dialogue agent utilizing persona and knowledge without the given candidates from the dataset. Due to the absence of knowledge candidates, the knowledge retriever retrieves the relevant paragraphs with the knowledge concept from the knowledge base. Also, the concept-based persona generator outputs the persona descriptions with the fragmentary persona concepts from retrieve-and-generate architecture. The generated persona descriptions are then validated through a persona aligner regarding relevancy and consistency. From experiments, we showed that our method is effective even though the persona concept and knowledge concept are given with the dialogue. We also presented the ablation studies on each component of our model. Moreover, we conducted the human evaluation to show the improved quality of the responses of our models and it is also shown in qualitative results. To show its applicability and adaptiveness, we denoted the experimental results of our method on FoCus, WoW, and PersonaChat datasets.

## 7 Acknowledgement

This research was supported by the MSIT(Ministry of Science and ICT), Korea, under the ITRC(Information Technology Research Center) support program(IITP-2023-2018-0-01405) supervised by the IITP(Institute for

Information & Communications Technology Planning & Evaluation). This work was supported by Institute of Information & communications Technology Planning & Evaluation(IITP) grant funded by the Korea government(MSIT) (No. 2020-0-00368, A Neural-Symbolic Model for Knowledge Acquisition and Inference Techniques). This research was supported by Basic Science Research Program through the National Research Foundation of Korea(NRF) funded by the Ministry of Education(NRF-2022R1A2C1007616)

## Limitations

Our model deal with response generation in the candidate-agnostic conversation setting, which is the limitation of INFO (Lim et al., 2022) model, proving the possibility of application in the real world. Still, hallucinations regarding personas and knowledge are observed occasionally in the generated responses. However, since the case of hallucinations is a severe problem even in large language models with enormous parameter sizes, it is required for our NLP communities to continue to solve the challenge. Also, although we conducted a human evaluation to validate the diverse capabilities of the proposed model, such as hallucination, consistency, and informativeness in dialogue generation, the number of cases is relatively small for evaluating the entire aspects of the capabilities. Finally, our model demands high GPU computation resources as it marginalizes loss at the token level.

We plan to improve our model for future work by conducting human evaluations with more cases and enhancing the way of qualitative analysis for the model's hallucinated answers. Improving the model's generator with more computationally efficient components is also a desirable direction for the GPU resource issues.

## Ethics Statement

We discuss the main ethical considerations of the model we proposed: (1) Privacy. the datasets adopted to construct our model provide factual knowledge and fictional person's preferences, and our model does not contain privacy issues. (2) Human evaluation. During the human evaluation process, we paid human workers the legal wage determined by the average time of evaluation and local labor compensation standards. We also guided them to take a rest when they are in a state

of fatigue during work. (3) Potential problems. Although we take conscientious steps to ensure the quality of our models, there can still be potential problems with the generated responses' quality, which can lead to incorrect predictions in applications that leverage factual information and human preferences. (4) Model deployment. Our approach employs the pre-trained language models (PLMs) for the downstream tasks, which have the risk of reflecting the bias of the training data. It is a well-known threat in tasks using PLMs, and we should be careful about social impact when using this method since our model aims to handle factual knowledge.

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

## A Training details for Concept-based Persona Generator

### A.1 Persona Retriever

For training persona retriever, we first construct persona pool. The statistics for persona pool is shown in Table 8. To train the persona retriever, we created a training and validation dataset using the persona pool. To construct a DPR-style dataset, we set the query as the persona-concept, the positive sample as the complete persona sentence, and the negative samples as lexically similar persona sentences retrieved from BM25 (Robertson and Zaragoza, 2009). We use haystack[3] and Hugging Face (Wolf et al., 2020) to implement persona retriever. The hyperparameters utilized for training persona retriever are batch size 32, learning rate $1 \times 10^{-6}$, AdamW optimizer (Loshchilov and Hutter, 2019) ($\beta_1 = 0.9$, $\beta_2 = 0.999$, and $\epsilon = 1e-8$). We initialized the query and context encoder with bert-base-uncased model and the training is carried out 5 epochs.

### A.2 Concept-based Persona Generator

For training the concept-based persona generator, we set top $k$ as 10 to augment ten relevant persona sentences to the input for training BART (Lewis et al., 2020a). The hyperparameters utilized for the training the BART are batch size 64, learning rate $5 \times 10^{-5}$, and the AdamW optimizer (Loshchilov and Hutter, 2019) ($\beta_1 = 0.9$, $\beta_2 = 0.999$, and $\epsilon = 1e - 8$). We initialized BART with facebook/bart-base from the Hugging Face and the training is carried out 5 epochs.

| Model | # Sentences |
|---|---|
| FoCus (Jang et al., 2022b) | 42,105 |
| PersonaChat (Zhang et al., 2018) | 4,795 |
| Total | 46,900 |

Table 8: Statistics for the persona pool used in persona retriever

## B Experimental Details

For the dialogue generation, we implement our method with Hugging Face (Wolf et al., 2020). We set the learning rate $6.25 \times 10^{-5}$ with the AdamW optimizer (Loshchilov and Hutter, 2019). We employ a $\text{BART}_{large}$ model with 12 encoder and decoder layers and RAG-Token that have the same number of layers for a fair comparison. The batch size is set as 128 for BART and 16 for RAG. $\lambda_P$ and $\lambda_{LM}$ are 0.5 and founded by manual search. The number of dialogue history window sizes is 2 and the beam size is 4. The whole model was trained for three epochs on RTX A6000 GPU and took 8 hours per epoch.

---

[3]https://github.com/deepset-ai/haystack

## C   Prompt for Building Aligner

```
Prompt
"Check for aligning" is whether the "Persona" is explicitly or implicitly reflected in "Dialogue".
Classify if a given "Dialogue" aligns with "Persona". You must answer with Yes or No.

New Sample

Persona: I would like to visit the Nazareth House again.
Dialogue:
A: I think I've been there before but I don't remember the name of this place.
B: This place is the Nazareth House, which you would like to visit again.
A: Can you describe this house to me?
B: You have curiosity about the description of Nazareth House and I will tell you. Nazareth House
is prominently located on an elevation along Wynnum North Road. The complex consists of a number
of buildings including the original building, the Convent and Chapel and two more recent additions,
St Joseph's Hostel and the nursing home known as Larmeniere.
A: Does this house look old to me, when it was built?
B: This house is relatively old, but since you would like to know when it was built, I will explain
it to you. Nazareth House was built from 1924 to 1939.
A: What is the history of this house?
B: The history of the house you are interested in began in 1925 when it was opened by Archbishop
James Duhig as part of the charity established on the site by the Poor Sisters of Nazareth. The
Nazareth House, located in Tingal Hill, Wynnum, was designed by the Brisbane architecture firm,
Hennessy, Hennessy, Keesing  Co and JP Donoghue and built by George Turner.

Check for aligning:

expected answer
{Yes or No}
```

Table 9: Prompt used for alignment on consistency by referring to persona and dialogue. Our persona consistency module is trained by gathering the responses.