# OpenReview forum: "Beyond Candidates : Adaptive Dialogue Agent Utilizing Persona and Knowledge"
_EMNLP/2023/Conference — EMNLP 2023 Findings_

### Official Review · Reviewer_wnZx · 2023-07-21

**Paper Topic And Main Contributions:** A dialogue adaptation method that use…
**Soundness:** 4

**Excitement:**

4: Strong: This paper deepens the understanding of some phenomenon or lowers the barriers to an existing research direction.

**Reasons To Accept:**

The authors present a simple, flexible, and effective method.

**Reasons To Reject:**

The method is somewhat ordinary and not impressively innovative

**Reproducibility:**

4: Could mostly reproduce the results, but there may be some variation because of sample variance or minor variations in their interpretation of the protocol or method.

**Reviewer Confidence:**

4: Quite sure. I tried to check the important points carefully. It's unlikely, though conceivable, that I missed something that should affect my ratings.

---

> ### Author Rebuttal · Authors · 2023-08-29
>
> We appreciate the reviewers’ constructive comments and suggestions. It is our belief that our paper will be improved to a great extent after following the comments. We will make our source code publicly available after the review process. Our response to the reviewers' comments is as follows: R, Q, and T indicate Reasons to Reject, Questions and Typos(+ Presentation), respectively.
>
> __[R1] The method is somewhat ordinary and not impressively innovative__
>
> Thanks for your considerate and constructive feedback. It is true that our main contribution is constructing persona and knowledge sentences from fragmentary information. However, we want to additionally emphasize our method’s extendability.
>
> 1) Extendability in Dataset:
>
> Our method can be applied regardless of the dataset type. In other words, our method is easily adapted to the dataset with a single source to ground. For instance, we can use the Wizard of Wikipedia dataset to implement persona-knowledge-grounded conversational agents by generating new personas. Also, we can build the agents with PersonaChat with an additional knowledge base as well. It implies that our methods can be utilized with any dataset grounding a single source.
>
> 2) Extendability in Model:
>
> Also, our method is capable of being utilized with any type of language model. Along with the simple knowledge retriever and persona aligner, it is extendible to any GLM.
>
> 3) Extendability in Services:
>
> Also, It can be applied to a service or application with ease. Suppose that users offer their preference tags to the personalized service. By just selecting few keywords instead of writing descriptive sentences, our persona generator can provide plausible and customized persona sentences with small information. Rather than simply retrieving persona sentences from a pre-defined persona pool, our persona generator can provide plausible and customized persona sentences without additional persona collection for retrieval. In fact, collecting the persona sentences from humans is costly and labor-intensive.

---

### Official Review · Reviewer_n8H1 · 2023-08-04

**Typos Grammar Style And Presentation Improvements:** 1. Lines 274-276 are not contiguous.
…
**Soundness:** 3

**Excitement:**

2: Mediocre: This paper makes marginal contributions (vs non-contemporaneous work), so I would rather not see it in the conference.

**Paper Topic And Main Contributions:**

The paper proposed a model to generate responses with fragmentary information, including persona and knowledge concepts.


**Questions For The Authors:**

1. In Section 3.3, why not just use the result of retrieval as personas?

**Reasons To Accept:**

1. The paper extended semantic-level conditional text generation to fragmentary information.

2. The evaluation metrics are sufficient.


**Reasons To Reject:**

1. The performance of the retriever was not evaluated.

2. The main contribution of this paper is only to construct sentence candidates from broken knowledge fragments through retrieval or generation.


**Reproducibility:**

4: Could mostly reproduce the results, but there may be some variation because of sample variance or minor variations in their interpretation of the protocol or method.

**Reviewer Confidence:**

4: Quite sure. I tried to check the important points carefully. It's unlikely, though conceivable, that I missed something that should affect my ratings.

---

> ### Author Rebuttal · Authors · 2023-08-29
>
> We appreciate the reviewers’ constructive comments and suggestions. It is our belief that our paper will be improved to a great extent after following the comments. We will make our source code publicly available after the review process. Our response to the reviewers' comments is as follows: R, Q, and T indicate Reasons to Reject, Questions and Typos(+ Presentation), respectively.
>
> __[R1] The performance of the retriever was not evaluated__
>
> We truly appreciate your valuable comments. We measure the retriever’s performance by checking whether the retrieved passages belong to the ground-truth knowledge source. The accuracy of the knowledge retriever is 75.19 %, and we will include the result in the final copy. It might be somewhat not enough compared to other retrievers in knowledge-grounded conversation. However, our work is the first time to show the build persona-knowledge grounding dialogue agents without candidates, so we focus on generating responses with little information. We will work on the improvements of our knowledge retriever performance in future work.
>
> __[R2] The main contribution of this paper is only to construct sentence candidates from broken knowledge fragments through retrieval or generation.__
>
> Thanks for your considerate and constructive feedback. It is true that our main contribution is constructing persona and knowledge sentences from fragmentary information. However, we want to additionally emphasize our method’s extendability.
>
> 1) Extendability in Dataset:
>
> Our method is extendible with respect to the type of dataset. Because our method is well applicable to the dataset with a single source to ground. For example, we can utilize the Wizard of Wikipedia dataset to implement persona-knowledge grounding dialogue agents by generating new personas. Also, we can build persona-knowledge grounding conversational agents with PersonaChat with additional knowledge base. It implies that our methods can be utilized with any dataset grounding single source.
>
> 2) Extendability in Model:
>
> Also, our method is flexible to utilize since our method can be applied regardless of the type of language model. Along with the simple knowledge retriever and persona aligner, it is extendible to any generative model.
>
> 3) Extendability in Services:
>
> Also, We emphasize that our method is extendible in services as well. Suppose that users offer their preference tags to the personalized service. By just selecting few keywords instead of writing descriptive sentences, our persona generator can provide plausible and customized persona sentences with small information. Rather than simply retrieving persona sentences from a pre-defined persona pool, our persona generator can provide plausible and customized persona sentences without additional persona collection for retrieval. In fact, collecting the persona sentences from humans is costly and labor-intensive.
>
> __[Q1] In Section 3.3, why not just use the result of retrieval as personas?__
>
> We are sorry for the insufficient explanation and we would like to clarify the situation of the training and inference phases. To cope with the candidate-free situation, we used pre-defined personas only in the training phase. However, unlike the training phases, only fragmentary information is used in the inference stage. As a result, our persona generator can manage situations where unseen and new persona sentences should be generated with fragmentary words. We hope this explanation is enough for your understanding and will add in the final copy if we get accepted.
>
> __[T1] Lines 274-276 are not contiguous.__
>
> Thank you for your detailed comments. We will revise the error following your suggestions.
>
> __[T2] It is recommended to use vector figures.__
>
> We really appreciate your feedbacks and will modify into the vector figures.

---

### Official Review · Reviewer_StNn · 2023-08-05

**Soundness:** 3

**Excitement:**

3: Ambivalent: It has merits (e.g., it reports state-of-the-art results, the idea is nice), but there are key weaknesses (e.g., it describes incremental work), and it can significantly benefit from another round of revision. However, I won't object to accepting it if my co-reviewers champion it.

**Paper Topic And Main Contributions:**

The paper focuses on persona / knowledge grounded response generation models. More specifically the authors try not to use pre-defined persona / knowledge candidates to help generate responses. They have a series of modules that retrieves a knowledge sentence using only a title and generate a persona using only certain concepts.

The main contribution as I see it is the proposed methodology to handle candidate free scenarios for persona / knowledge grounded dialog.

**Reasons To Accept:**

(1) The motivation behind this work is strong as it can be cumbersome to keep a set of pre-defined candidates for personas and the idea of generating one during the conversation is great.

(2) There is an extensive set of experiments done.

**Reasons To Reject:**

(1) It is not clear how the persona generator works. It seems that pre-defined personas are still being used in which case what is novel about this?


(2) 25 dialogs for human evaluation seems too small. Additionally the strongest baseline model that was shown in automatic evaluation appears to be missing in human evaluation.

**Reproducibility:**

3: Could reproduce the results with some difficulty. The settings of parameters are underspecified or subjectively determined; the training/evaluation data are not widely available.

**Reviewer Confidence:**

4: Quite sure. I tried to check the important points carefully. It's unlikely, though conceivable, that I missed something that should affect my ratings.

**Typos Grammar Style And Presentation Improvements:**

It would be good to formally define what is a knowledge / persona concept and how that differs from a knowledge / persona candidate.

There is some unusual formatting in the paper such as large gaps in between paragraphs.

---

> ### Author Rebuttal · Authors · 2023-08-29
>
> We appreciate the reviewers’ constructive comments and suggestions. It is our belief that our paper will be improved to a great extent after following the comments. We will make our source code publicly available after the review process. Our response to the reviewers' comments is as follows: R, Q, and T indicate Reasons to Reject, Questions and Typos(+ Presentation), respectively.
>
> __[R1] It is not clear how the persona generator works. It seems that pre-defined personas are still being used in which case what is novel about this?__
>
> Thanks for your considerate and constructive feedback. We agree with your concerns, and we would like to clarify the situation of the training and inference phases. To cope with the candidate-free situation, we used pre-defined personas only in the training phase. However, unlike the training phases, only fragmentary information is used in the inference stage. As a result, our persona generator can manage situations where unseen and new persona sentences should be generated with fragmentary words. In detail, our persona generator is pre-trained to generate plausible full persona descriptions with only little information (persona concept) in a retrieve-and-generate manner.  Then, we freeze the persona generator for the response generation. More details are indicated in Section 3.3
>
> Also, the persona generator in the inference stage is extendible and less labor-intensive. Suppose that users offer their preference tags to the personalized service. By just selecting few keywords instead of writing descriptive sentences, our persona generator can provide plausible and customized persona sentences with small information. In other words, retrieving persona sentences from a pre-defined persona pool and additional persona collection for retrieval, which is costly and labor-intensive, are unnecessary with our method. Therefore, our method is capable of applying candidate-free situated dialogue grounding persona and knowledge with ease.
>
> __[R2] 25 dialogs for human evaluation seems too small. Additionally, the strongest baseline model that was shown in automatic evaluation appears to be missing in human evaluation.__
>
> Thanks for your valuable comments. Since it is a high expense to have a human evaluation, we only conduct the evaluation for 25 dialogues within our budget. We are additionally conducting 100 more human evaluations for its validity. We will include the human evaluation results for the final copy if the paper is accepted. Moreover, we did not include the INFO’s results in the human evaluation because we aim to evaluate the response generation results only with the candidate-free situated model. We hope that our evaluation setting is convincing with this explanation.
>
> __[T1] It would be good to formally define what is a knowledge / persona concept and how that differs from a knowledge / persona candidate.__
>
> Thanks for your detailed comments. In addition to the explanation of the difference between candidates and concepts in Figure 1, we will additionally emphasize the difference between knowledge and persona concepts with knowledge and persona candidates, along with notation and explanation.
>
> __[T2] There is some unusual formatting in the paper such as large gaps in between paragraphs.__
>
> We will reallocate the Tables and Figures to lessen the gaps between the paragraphs. Thanks for the considerate feedback.

---

### Meta-Review · Area_Chair_NJQh · 2023-09-25

**Recommendation:** 3

**Metareview:**

The paper proposes a persona-grounded dialog generation pipeline. The paper lacks extendability and generalizability beyond what can be hypothesized (author responses are less convincing). Hence, the contribution of the paper is a bit limited.

---

### Meta-Review · Senior_Area_Chairs · 2023-10-05

**Recommendation:** 3

**Metareview:**

meta review

---

### Decision · Program_Chairs · 2023-10-07

**Decision:**

Accept-Findings

**Comment:**

The paper proposes a persona-grounded dialog generation pipeline. The paper lacks extendability and generalizability beyond what can be hypothesized (author responses are less convincing). Hence, the contribution of the paper is a bit limited.|meta review